# Propolis as a Potential Disease-Modifying Strategy in Parkinson’s disease: Cardioprotective and Neuroprotective Effects in the 6-OHDA Rat Model

**DOI:** 10.3390/nu12061551

**Published:** 2020-05-26

**Authors:** Valeria C. Gonçalves, Daniel J. L. L. Pinheiro, Tomás de la Rosa, Antônio-Carlos G. de Almeida, Fúlvio A. Scorza, Carla A. Scorza

**Affiliations:** 1Disciplina de Neurociência, Departamento de Neurologia e Neurocirurgia, Universidade Federal de São Paulo (UNIFESP), São Paulo 04039-032, Brazil; pdanielleal@gmail.com (D.J.L.L.P.); tomas.rosa@unifesp.br (T.d.l.R.); scorza.nexp@epm.br (F.A.S.); 2Laboratório de Neurociências Experimental e Computacional, Departamento de Engenharia de Biossistemas, Universidade Federal de São João del-Rei (UFSJ), Minas Gerais 36301-160, Brazil; acga@ufsj.edu.br

**Keywords:** Parkinson’s disease, propolis, 6-OHDA

## Abstract

Patients with Parkinson’s disease (PD) manifest nonmotor and motor symptoms. Autonomic cardiovascular dysregulation is a common nonmotor manifestation associated with increased morbimortality. Conventional clinical treatment alleviates motor signs but does not change disease progression and fails in handling nonmotor features. Nutrition is a key modifiable determinant of chronic disease. This study aimed to assess the effects of propolis on cardiological features, heart rate (HR) and heart rate variability (HRV) and on nigrostriatal dopaminergic damage, detected by tyrosine hydroxylase (TH) immunoreactivity, in the 6-hydroxydopamine (6-OHDA) rat model of PD. Male Wistar rats were injected bilaterally with 6-OHDA or saline into the striatum and were treated with propolis or water for 40 days. Autonomic function was assessed by time domain parameters (standard deviation of all normal-to-normal intervals (SDNN) and square root of the mean of the squared differences between adjacent normal RR intervals (RMSSD)) of HRV calculated from electrocardiogram recordings. Reductions in HR (*p* = 1.47 × 10^−19^), SDNN (*p* = 3.42 × 10^−10^) and RMSSD (*p* = 8.2 × 10^−6^) detected in parkinsonian rats were reverted by propolis. Propolis attenuated neuronal loss in the substantia nigra (*p* = 5.66 × 10^−15^) and reduced striatal fiber degeneration (*p* = 7.4 × 10^−5^) in 6-OHDA-injured rats, which also showed significant weight gain (*p* = 1.07 × 10^−5^) in comparison to 6-OHDA-lesioned counterparts. Propolis confers cardioprotection and neuroprotection in the 6-OHDA rat model of PD.

## 1. Introduction

Parkinson’s disease (PD) is a frequent and debilitating age-related neurodegenerative disease associated with health deterioration and augmented risk of mortality [1,2]. The number of PD subjects is estimated to increase considerably in the years to come due to population ageing and rise in global average life expectancy [3,4]. Although the core motor clinical manifestations of PD resulting from the progressive loss of nigrostriatal dopaminergic signaling are the pathological emblem of the disease, PD is accompanied by a constellation of nonmotor features that lead to substantial burden for PD patients and impair their quality of life [5,6]. Autonomic dysfunctions have incited great clinical interest since they frequently appear as PD nonmotor symptoms, afflicting up to 90% of PD individuals and imposing considerable burden in terms of morbimortality [7]. Autonomic dysfunctions may precede the onset of motor symptoms, and signs of cardiovascular dysautonomia commonly occur in PD patients [8,9]. Recently, a prospective epidemiological study revealed the association between reduced HRV measures in subjects without PD and augmented risk of developing the disease [10]. Many reports have identified cardiovascular changes in PD patients, represented by a decrease in heart rate (HR) and heart rate variability (HRV) indices (standard deviation of all normal-to-normal intervals (SDNN) and square root of the mean of the squared differences between adjacent normal RR intervals (RMSSD)), related to an increase in mortality in these individuals [11,12]. Research has shown that HRV can decline significantly with compromised health status, suggesting that HRV can be considered as a biological indicator of cumulative allostatic load [13,14]. Although the link between low HRV and increased morbidity has been reported in PD patients, little has been written regarding the use of animal models of PD in the screening of cardiovascular dysfunction associated with neurodegeneration of the nigrostriatal dopaminergic system. A widely adopted rodent model of PD is induced by injections of 6-hydroxydopamine (6-OHDA) into the striatum, resulting in progressive degeneration of the dopaminergic neurons in the nigrostriatal tract [15,16]. In addition to impaired voluntary movements, 6-OHDA-injured rats exhibit a variety of nonmotor disturbances [17]. Some studies have shown that 6-OHDA-treated rats have lower patterns of HR and HRV than their control counterparts [18,19,20]. Knowledge of the pathological dysfunctions in an animal model is critical for recognizing prospective therapeutic targets as adjunctive therapies aimed at multiple levels of pathology are required in PD. The etiology of PD remains elusive. Nevertheless, it appears linked to mitochondrial dysfunction, immune responses, oxidative stress, and inflammation, leading researchers to search for strategies to dampen these detrimental processes [21,22,23,24]. There is no clinical intervention to slow down or cease disease progression. Hence, new treatment options for alleviating disease impact remain an unmet need in PD. Environmental factors seem to exert a pivotal role in PD progression and, among them, lifestyle-related factors, such as nutrition, have been widely studied areas due to their potential beneficial roles in the management of PD [24,25]. Natural products are potential candidates for novel adjunctive therapeutic approaches and have gained prominence in the literature, such as ginseng [26,27], curcumin [28,29], cannabidiol [30,31], green/black tea [32,33], and coffee [34,35]. Among the promising natural products, Brazilian propolis is an important source of a variety of bioactive components with reported biological and pharmacological actions, such as anti-inflammatory, anesthetic, antioxidant, and immunomodulator, among others, attributed to a composition rich in caffeic, p-coumaric, and cinnamic acids [36,37,38,39]. The chemical properties of propolis can vary from one sample to another as its constitution derives from species of bees and the availability of plant sources [40,41]. It is composed of resinous and balsamic substances (50%–60%), waxes (30%–40%), essential oils (5%–10%), and pollen grains (5%) associated with microelements (calcium, strontium, iron, copper, aluminum, manganese, and vitamins B1, B2, B6, C, and E). These are collected from plants by bees and mixed with their salivary secretions, resulting in a yellowish substance used in the construction, maintenance, and protection of hives, limiting the growth of bacteria and fungi [42,43]. Research on animal models and human cells has suggested the beneficial effects of propolis on coronary heart diseases and risk factors associated with cardiovascular diseases as well as on acute and chronic brain processes [44,45,46,47,48,49,50,51]. As disease-modifying approaches remain a crucial goal in PD, we aimed to examine whether propolis can alleviate heart rate parameters and nigrostriatal dopaminergic loss in the 6-OHDA model of PD. In addition, as managing unintentional weight loss in patients seems to be a major element of PD, the effect of propolis on the weight of rats was assessed.

## 2. Materials and Methods

### 2.1. Animals

In this study, we used 44 adult male Wistar rats, weighing 230–300 g and aged 8 weeks old. The animals were purchased from the central bioterium of Federal University of São Paulo and were kept in the animal bioterium of the neuroscience laboratory, housed in groups of 4 per cage with appropriate sawdust and free access to water and food under a light–dark cycle (light: 7:00–19:00) and constant temperature of 21 ± 2 °C. All procedures were performed according to the guidelines established by the Ethics Committee of the Federal University of São Paulo (CEUA number 4512290118), and every effort was made to minimize animal suffering.

### 2.2. Study Design

The rats were distributed into groups of the same size as follows: sham, sham + P, 6-OHDA, and 6-OHDA + P. The 6-OHDA animals received bilateral injection of 6-hydroxydopamine neurotoxin into the striatum according to the stereotactic coordinates described in the Paxinos and Watson rat brain atlas [52]. Sham animals received saline injection at the same brain points. Twenty-four hours after stereotaxic surgery, animals in the groups with the +P termination received daily gavages with Brazilian green propolis (200 mg/kg) [53,54,55], while rats in the other groups received filtered water for 40 days. Rats were weighed once a week. On the 30th day after surgery, cardiac electrodes were implanted in the animals, and electrocardiogram recordings were carried out from day 35 up to day 40. The electrocardiogram (ECG) recordings of the 40th day were used for data analysis, and the previous recordings were intended for the animals’ adaptation to the experimental condition. At the end of experiments, the animals were euthanized by transcardiac perfusion, and the brains were removed for immunohistochemistry.

### 2.3. 6-OHDA Lesion

The 6-OHDA model is a classic and widely used neurotoxin rodent model of PD. The bilateral administration of 6-OHDA into the striatum induces retrograde neuronal cell death of the tyrosine hydroxylase (TH)-containing neurons in the substantia nigra pars compacta (SNc) mimicking Parkinson-like pathology [56,57]. Here, the animals were anesthetized with ketamine (100 mg/kg) and xylazine (10 mg/kg) intraperitoneally and placed in the stereotactic device (EFF 331-Insight™, Ribeirão Preto, São Paulo, Brazil) (Figure 1A). The 10 µL Hamilton syringe attached to the stereotactic rod was used to inject 1 µL of 6-OHDA (Sigma^©^, Saint Louis, Missouri, USA) solution (12 µg/µL concentration in 0.3% ascorbic acid) or saline solution in four different coordinates: (1) laterolateral: −2.7 mm, anteroposterior: bregma, dorsoventral: −4.5 mm; (2) laterolateral: −3.2 mm, anteroposterior: +0.5 mm, dorsoventral: −4.5 mm; (3) laterolateral: +2.7 mm, anteroposterior: bregma, dorsoventral: −4.5 mm; and (4) laterolateral: +3.2 mm, anteroposterior: +0.5 mm, dorsoventral: −4.5 mm (Figure 1B,C).

### 2.4. Implantation of Cardiac Electrodes and Electrocardiogram

The cardiac electrode was built by us in the laboratory from stainless steel wires coated with nylon and welded to an output connector. Thirty days after the injury induced by 6-OHDA, the electrode was positioned above the skull cap of the animal with the aid of self-curing acrylic resin. The cardiac electrode wires were subcutaneously guided to the xiphoid process and to the central region of the sternocleidomastoid muscles (Figure 2A,B). After recovery, ECG recordings were performed from day 35 to day 40. The animals in waking state were submitted to electrocardiographic records at a sampling frequency of 1000 Hz (Figure 2C). The signals were recorded and transduced by an amplification circuit and then digitized by the PowerLab v8 device (AD Instruments, Australia) and transferred to be processed in the LabChart v8 software (AD Instruments, Australia). HR and HRV parameters were evaluated based on the distance between the R-R intervals. The calculations were performed using the MATLAB program on an 8 GB RAM computer with an Intel^®^ Core™ i7-6700 processor, 3.4 GHz. Eight stretches of one minute were selected from different stationary sections of the electrocardiographic signals. Afterwards, all R-R peaks were identified by the amplitude greater than the signal mean plus 70% of its standard deviation and minimum distance between peaks of 100 ms (Figure 2D). Based on the interval between the R-R peaks, the tachogram was calculated, a time series with the intervals between the R-R peaks (Figure 2E). From the tachogram, it was possible to calculate the HR and HRV parameters (SDNN and RMSSD) (Figure 2F) as well as the histograms of the distributions of these intervals for each experimental group and the statistical moments related to them.

### 2.5. Propolis Gavage

On the first day after the 6-OHDA injection, propolis solution was administered at a dose of 200 mg/kg [53,54,55] to the sham + P and 6-OHDA + P animals and filtered water to the others until the 40th day. The standardized propolis extract (EPP-AF^®^, Ribeirão Preto, São Paulo, Brazil) employed in this study was kindly provided by Apis Flora Co. and is a composition containing mostly green propolis in water-soluble dry extract form [58]. The EPP-AF^®^ is a chemically and biologically reproducible pharmaceutical compound.

### 2.6. Animal Weighing

All animals were weighed weekly on a balance before gavage to assess possible weight changes during experimental procedures.

### 2.7. Immunohistochemistry and Neuronal Death Analysis

Animals were euthanized by transcardiac perfusion after intraperitoneal injection of lidocaine (10 mg/kg) and sodium thiopental (80 mg/kg). The rib cage was opened, a needle in the left ventricle was introduced, and 0.01 M phosphate buffered saline (PBS) (pH 7.4) and 4% paraformaldehyde (PFA) (pH 7.4) were injected (Figure 3A,B). Solutions were drained by an incision in the right atrium. The brain was removed and placed in 4% PFA for 24 h and then immersed in 30% cryoprotective sucrose solution for 24 h to be cut (coronal sections of 40 μΜ) using a cryostat (Microm HM 505E) (Figure 3C,D). Briefly, slices were washed three times in 0.01 M PBS (pH 7.4), treated with 0.1% hydrogen peroxide and, after washing, incubated in 10% albumin solution and 0.3% Triton X-100 for 2 h. Slices were incubated overnight with primary antibody (1:1000 tyrosine hydroxylase, Abcam^®^) diluted in 0.01 M PBS (pH 7.4) and 2% albumin. Dopaminergic neurons were identified by TH-positive immunoreactivity. After three washes with 0.01 M PBS (pH 7.4), slices were treated with biotinylated secondary antibody (anti-rabbit 1:200-Abcam^®^) diluted in 0.01 M PBS and 2% albumin, washed and incubated with avidinbiotin–peroxidase complex (ABC Elite; Vector Labs, Burlingame, CA, USA). Next, sections were stained with 3,3′-diaminobenzidine (DAB) tetrachloride dissolved in 0.05 M TRIS–HCl (PH 7.6) activated by 0.3% hydrogen peroxide. Finally, slices were placed on silanized slides, subjected to a dehydration process and diaphanization (Figure 3E), and the slides were photographed using a 4× zoom optical microscope (Nikon ECLIPSE E600). The images were analyzed (TH-positive immunoreactive neurons in the SNc and striatal fibers) by the ImageJ program. The TH-immunopositive striatal fiber densities were assessed in both hemispheres in five brain sections of 40 µm for each area by calculating the optical density differences between different groups versus the control group. For the SNc regions, TH-stained neural nuclei were counted.

### 2.8. Statistics

The Kolmogorov–Smirnov test was used to test the normality of data regarding animal body weight, immunohistochemistry, and parameters from the ECG recordings, which were not derived from a normal distribution. Thus, the nonparametric Kruskal–Wallis test was used to simultaneously compare the four groups studied. The Bonferroni post hoc test was performed to determine which groups were significantly different from the others. The level of statistical significance was set at 0.05. All statistical tests were performed using the MATLAB R2017a.

## 3. Results

### 3.1. Analyses of Heart Rate and Heart Rate Variability

The 21 animals (groups: sham, *n* = 6; sham + P, *n* = 5; 6-OHDA, *n* = 5; 6-OHDA + P, *n* = 5) were registered from the 35th to the 40th day, and the analyses were performed using ECG signals recorded on day 40 (Figure 4). The results showed a significant difference between all groups for HR parameter when compared to 6-OHDA group and to 6-OHDA + P group (chi-square = 90.79, *p* = 1.47 × 10^−19^) (Figure 5A). Biological systems show patterns of variability, and HRV describes the oscillations between heartbeats. Through analyses of SDNN, a measurement of HRV in the time domain, a significant difference was found between all groups and the 6-OHDA group and between the groups sham + P and 6-OHDA + P (chi-square = 47.03, *p* = 3.42 × 10^−10^) (Figure 5B). For the RMSSD, another index in the time domain for the HRV analysis, significant differences were obtained in all groups when compared to the 6-OHDA group of rats (chi-square = 26.31, *p* = 8.2 × 10^−6^) (Figure 5C). For the analysis of HR, SDNN, and RMSSD, excerpts from the ECG signals of each animal were used to quantify the R-R peaks. The HR values of the 6-OHDA-lesioned rats were significantly lower than those of the sham animals. However, the 6-OHDA animals treated with propolis showed significantly higher HR in comparison to 6-OHDA-lesioned rats treated with filtered water. HRV analyses in time domain give information about autonomic activity. Our results showed that all groups exhibited higher HRV, taking in consideration the parameters measured over time, when compared to the 6-OHDA group treated with water, suggesting impaired autonomic cardiovascular regulation in rats with Parkinson-like lesion that received water. The results suggest that propolis markedly mitigated the deleterious cardiovascular effects associated with parkinsonism induced by 6-OHDA. From the distribution of the intervals between R-R peaks, intersections between some classes of the histogram were observed (Figure 6), which were detected by the proximity of the mean and standard deviation (Table 1). The histogram can provide a broad idea about the shape of data distribution. However, skewness and kurtosis are two numerical shape measurements that offer a more precise interpretation, and both were applied here in the analyses related to the shape of the distributions. Skewness is a symmetry measurement, meaning that the value of a symmetrical distribution is zero; positive skewness (greater than 1) indicates that the right-side tail is longer, while a negative value (less than 1) means the left-side tail is longer in comparison to the right one. The kurtosis analysis assesses the flattening of the histogram curve. Distributions with kurtosis greater than 3 mean that there is greater weight of the data distribution to one side of the curve, that is, there is a large amount of data above or below the average. Thus, the propolis-treated 6-ODHA group showed a tendency to have shorter-than-average distances between R-R peaks than the water-treated 6-OHDA group, that is, 6-OHDA rats treated with propolis tended to have an HR above average when compared to other experimental groups.

### 3.2. Tyrosine Hydroxylase Immunohistochemistry

The immunohistochemistry procedure was performed on 17 animals (groups: sham, *n* = 3; sham + P, *n* = 4; 6-OHDA, *n* = 5; 6-OHDA + P, *n* = 5). TH-positive immunoreactivity was applied as a marker of viable dopaminergic neurons in the SNc and dopamine fibers in the striatum (Figure 7 and Figure 8). In the striatum, an optical density analysis was carried out, and it showed that all groups were significantly different from the 6-OHDA group (chi-square = 22.51, *p* = 5.11 × 10^−5^) (Figure 9A). In the SNc region, by counting dopaminergic neuronal nuclei, significant differences were detected in all groups when compared to both 6-OHDA group and 6-OHDA + P group (chi-square = 45.72, *p* = 6.5016 × 10^−10^) (Figure 9B). Nevertheless, 6-OHDA-injured rats exhibited less TH-stained striatal fibers and less TH-positive neural nuclei in the SNc than the 6-OHDA-lesioned rats treated with propolis, suggesting a protective effect of propolis against 6-OHDA-induced neurodegeneration of the dopaminergic nigrostriatal pathway. Our results showed that 6-OHDA injection resulted in an average reduction of 53.38% in the dopaminergic neuronal nuclei in the SNc area and an average reduction of 25% in the striatal fibers compared to the control group. In contrast, 6-OHDA rats treated with propolis showed an average neuronal loss of 28.12% in the SN and an average decrease of 17.7% in the striatal fiber density compared to the control group.

### 3.3. Animal Weight

All animals were weighed once weekly during the experimental period of 40 days (groups: sham, *n* = 13; sham + P, *n* = 13; 6-OHDA, *n* = 9; 6-OHDA + P, *n* = 9) (Figure 10A). The analysis of the body weight evolution showed no statistically significant differences between groups on day 0 (chi-square = 2.2876, *p* = 0.5149) (Figure 10B). However, a significant difference between all groups was found on day 40 (chi-square = 25.7568, *p* = 0.0000107), except between the sham group and the sham + P group (Figure 10C). The 6-OHDA-lesioned rats presented lower weight gain in comparison to the saline-injected rats. However, the 6-OHDA-injured rats treated with propolis gained significantly more weight than the 6-OHDA-injured animals treated with water.

## 4. Discussion

Parkinson’s disease is an incurable, complex, and slowly progressive neurological disorder with a wide variety of motor and nonmotor signs [59]. Among nervous system pathologies surveyed in the Global Burden of Disease study [60], PD was found to be “the fastest growing in prevalence, disability, and deaths” [61]. Current disease treatments aim to control the symptomatic manifestations of PD but do not interrupt or slow its progression, and although dopamine replacement therapy remains the main treatment for the motor symptoms of PD, its long-term clinical ineffectiveness demands new approaches capable of increasing the survival of nigrostriatal dopaminergic neurons [62,63,64]. Traditional clinical treatments fail in handling the additional disruptive signs of PD, such as cardiovascular dysfunctions, which can be expected to cause severe complications, including death. As a result, novel add-on therapies for PD are needed. Here, we showed that propolis conferred protection against loss of SNc dopaminergic neurons and striatal fibers in the 6-OHDA rat model of PD. In addition, propolis had beneficial cardiovascular effects in rats treated with 6-OHDA and relieved unwanted weight loss in animals with PD.

PD is characterized by progressive death of dopamine-containing neurons in the SNc. The striatal injection of the neurotoxin 6-OHDA leads to a progressive nigrostriatal dopaminergic pathway degeneration, mimicking human PD [65,66]. In our work, propolis mitigated 6-OHDA-induced loss of TH-positive nigrostriatal dopaminergic neurons. Yet, despite the many variables that can hinder the comparison between different studies in animals evaluated by independent groups (type of model, methodological approaches, propolis composition, supplementation period, and concentration and dosage, among others), evidence has suggested propolis and its main flavonoids as neuroprotective agents, preserving brain tissues against many forms of neurotoxicity [67,68,69,70,71,72,73]. Barros Silva and collaborators [74] described a decrease in dopaminergic loss triggered by unilateral injection of 6-OHDA in the SNc and striatum of rats after administration of caffeic acid, a flavonoid present in Brazilian green propolis. The authors of [74] suggested the elimination of both free radicals and reactive oxygen species as the underlying mechanisms of the antioxidant properties responsible for the neuroprotective effects of propolis. Oxidative stress is recognized as a crucial contributor of dopaminergic cell death in the SNc in genetic and sporadic forms of PD [75]. Clinical studies have suggested the increased presence of both reactive oxygen/nitrogen species and neuroinflammatory molecules in the serum of PD patients as major collaborators of the pathophysiology of disease [76]. Augmented levels of oxidative stress markers and reduced antioxidant capacity have been found in the liquor of healthy individuals with PD-associated LRRK2 mutations, the most common cause of inherited PD [77]. Caffeic acid, a component found in propolis, was shown to decrease brain inflammatory markers and protect nigral dopaminergic neurons in the rotenone-induced mouse model of PD [78]. Similarly, the neuroprotective roles of chrysin, a flavonoid encountered in propolis, were achieved through mechanisms involving inflammatory mediators and neurotrophic factors in the unilateral 6-OHDA model of PD [73]. Cinnamic acid, another component present in green propolis, showed neuroprotective effects in the 1-methyl-4-phenyl-1,2,3,6-tetrahydropyridine (MPTP) mouse model of PD, detected by the attenuation of the nigrostriatal cell loss via activation of peroxisome proliferator-activated receptor alpha, which is involved in regulation of energy homeostasis and is known to have anti-inflammatory and antioxidative stress properties [79]. The existing literature has suggested important biological activities of propolis, including its antioxidant, anti-inflammatory, analgesic, and antimicrobial properties, which may contribute to its neuroprotective and cardioprotective roles [69,80].

There is still a huge gap in knowledge about the effects of propolis on cardiovascular parameters in animal models of PD. Reduced HRV attributed to autonomic nervous system dysfunction is often detected in PD patients, increasing morbidity in these individuals [23,81,82]. Studies using PD animal models have raised critical aspects regarding cardiovascular complications in Parkinson-like disease pathophysiology [30,32,83,84,85]. Studies on the 6-OHDA model of PD have shown the cardiovascular effects associated with dopaminergic neurodegeneration, suggesting the modulation of cardiovascular parameters by a central pathway [30,84,86,87,88,89,90,91]. In line with our results, Ariza [31] reported bradycardia in 6-OHDA-lesioned rats only seven days after bilateral injections in the SNc, which resulted in rapid onset of neuronal death. In a previous work by our group, we described the reduction in both heart rate and HRV in rats with parkinsonism at 15 days following unilateral intrastriatal delivery of 6-OHDA [32]. The 6-OHDA-intrastriatal delivery model of PD mimics the progressive degenerative nature of the human disease and is an ideal condition for studying disease-modifying treatment strategies [92,93]. As far as we know, this is the first work to assess the effects of propolis on cardiovascular parameters in an animal model of PD. We showed that propolis promoted the increase of both HR and HRV parameters in 6-OHDA-lesioned rats, suggesting its beneficial cardiovascular effects. Parkinsonian rats that received propolis showed a decrease in time between the R-R peaks (150–212 ms) in comparison to parkinsonian rats treated with water (175–237 ms), which means increased HR parameters linked to propolis consumption. In the evaluation of HRV measurements, the RMSSD index is related to parasympathetic activity and allows short-term HR analysis by identifying abrupt changes between the R-R intervals [94]. SDNN is used to assess both sympathetic and parasympathetic activity on the heart, but it is not possible to distinguish which of the two activities influences HRV changes the most. Our results obtained from SDNN analysis indicated increased values in the 6-OHDA-lesioned rats treated with propolis when compared with 6-OHDA-injured animals treated with water. Taken together, the HRV measurements suggest that propolis has a cardioprotective effect on the 6-OHDA rat model of PD. Previous studies have shown statistically significant reductions in frequency domain and time domain measures (RMSSD and SDNN) of HRV, suggesting compromised autonomic cardiovascular regulation in PD patients, which is associated with greater risk of mortality [95,96,97,98]. Studies have suggested that HRV analysis is a major approach for assessing early sign of autonomic dysfunction in patients with PD and is more accurate than the conventional cardiovascular tests that are commonly used, such as Valsalva maneuver and head-up tilt test, among others [22,24,99]. In addition to human PD, diminished HRV obtained from analyses of ambulatory ECG recordings has been described in several diseases, such as epilepsy, stroke, systemic lupus erythematosus, diabetes mellitus, and myocardiopathy, among others. It has been suggested that decreased HRV measurements in patients with acute and chronic heart diseases are associated with subclinical inflammation and that plasma concentrations of inflammatory markers are inversely correlated with HRV [100,101,102]. Systemic injection of the catecholaminergic toxin 6-OHDA mimics the PD-like cardiac sympathetic neurodegeneration associated with augmented inflammation and oxidative stress in nonhuman primates [86]. We have previously shown that narrower heart rate in 6-OHDA-injured animals is associated with increased heart levels of inflammatory factors and decreased concentrations of vitamin D, a regulator of immune responses [85]. Results from studies using the conventional 6-OHDA model of PD highlight the intricate role of the nigrostriatal dopaminergic loss in cardiac dysautonomia [30,32,83,85,103,104,105]. At the time of diagnosis, approximately 60% of PD subjects exhibit loss of cardiac sympathetic innervations that cannot be prevented or slowed and, due to its progressive nature, it ultimately reaches 100% of PD patients. However, symptoms are not alleviated by current antiparkinsonian drug therapy [106]. Taken together, our results suggest propolis as a potential natural product to fight against central dopamine cell depletion and cardiovascular dysfunction in PD and as a potential candidate for the development of new adjunctive therapies.

PD subjects may start to lose weight years prior to motor symptoms. Unintentional weight loss attributed to PD is frequent and linked to worse quality of life in patients [107,108]. Reduced calorie consumption and increased energy expenditure have been suggested as the undesirable cause of weight loss in PD [109,110,111]. A recent study assessed the association between dysautonomia and body weight in PD subjects by the investigation of parasympathetic dysfunction (HRV measurement and the existence of constipation) and sympathetic denervation (orthostatic hypotension and cardiac uptake of ^123^I–metaiodobenzylguanidine (^123^I-MIBG)) [110]. The authors found that lower HRV in underweight PD patients was associated with greater disease severity, diminished blood pressure in tilt table test, enhanced cardiac washout ratio of ^123^I-MIBG, and more frequent constipation complaints in comparison to normal weight or overweight PD subjects. Therefore, the authors concluded that autonomic nervous system dysfunction appears to be closely related to weight loss in PD [110]. Guimarães and colleagues [112] used the bilateral 6-OHDA rat model of PD to study the effects of locus coeruleus noradrenergic neurodegeneration on body weight loss in PD-like disease. In addition to bilateral 6-OHDA striatal lesions, parkinsonian rats received additional 6-OHDA injuries in the locus coeruleus, a brain area that plays a role in the regulation of cardiovascular activity, leading to significant weight loss that was reverted by deep brain stimulation of the subthalamic nucleus [112]. In the present study, the parkinsonian rats had significant weight loss that was reverted by propolis consumption. This is the first study to show the effects of propolis against weight loss in an animal model of PD.

In conclusion, Brazilian green propolis showed neuroprotective effects against nigrostriatal dopaminergic loss and cardioprotective effects against cardiac autonomic dysfunction in the bilateral 6-OHDA model of PD. In addition, propolis abolished the undesirable body weight loss associated with PD. Studies aimed at understanding the underlying biological mechanisms of actions of propolis remain to be conducted. Our findings suggest propolis as a potential candidate for the development of an adjunctive therapy for counteracting motor and nonmotor symptoms of PD.

## Figures and Tables

**Figure 1 nutrients-12-01551-f001:**
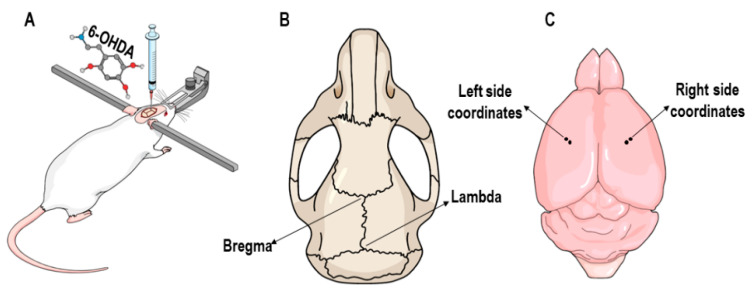
Schematic figure of the 6-hydroxydopamine (6-OHDA)-induced bilateral striatal lesion. (**A**) Animal fixed to stereotactic apparatus receiving injection of the neurotoxin by Hamilton syringe. (**B**) Rat skull, the arrows indicate the anthropometric points (lambda and bregma). (**C**) Rat brain, the arrows indicate the points referring to the stereotactic coordinates used for bilateral injection of 6-OHDA into the striatum. Picture created on the Mind the Graph platform (mindthegraph.com) under free license.

**Figure 2 nutrients-12-01551-f002:**
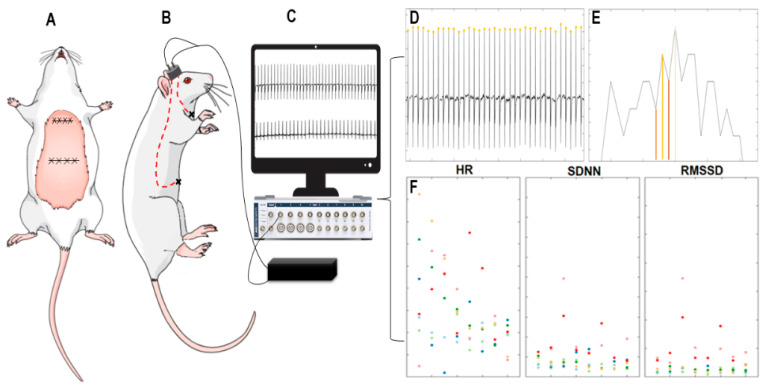
Schematic figure of the cardiac lead implantation and signal analysis. (**A**) Suture indicating the location of the xiphoid process and sternocleidomastoid musculature. (**B**) The acrylic structure in the rat skullcap with the cardiac electrode output connected to the electrocardiogram (ECG) register acquisition equipment. The red lines indicate the cardiac electrode wires placed subcutaneously. (**C**) Cardiac lead output connected to the biological signal amplifier with output for PowerLab v8 data digitizer and representative electrocardiographic record. (**D**) Identification of the amplitude of each R point of the analyzed stretch represented by the yellow dot. (**E**) Tachograph of R-R points. Vertical lines identify the R peaks. The distance between the lines on the x-axis represents the distance R-R. The height, on the y-axis, represents the maximum amplitude of the R peak. (**F**) Average heart rate, standard deviation of all normal-to-normal intervals (SDNN), and square root of the mean of the squared differences between adjacent normal RR intervals (RMSSD) dispersion of each stretch analyzed by group of animals. Picture created on the Mind the Graph platform (mindthegraph.com) under free license.

**Figure 3 nutrients-12-01551-f003:**
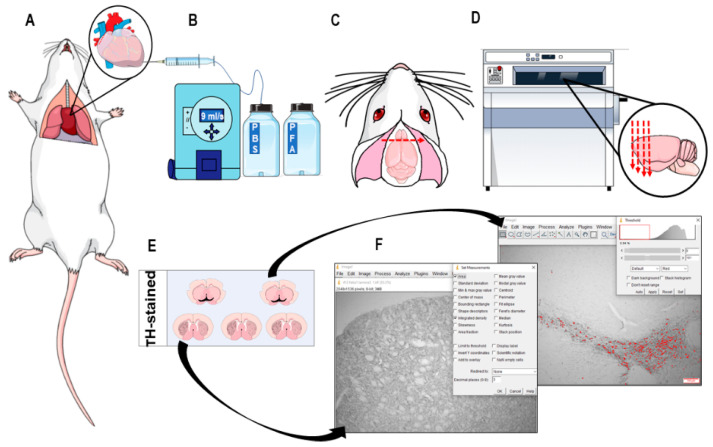
Representative schematic of the major experimental steps to assess nigrostriatal dopaminergic death. (**A**) Transcardiac perfusion: animal with exposed rib cage. The insert shows the heart with a cut in the right atrium for drainage of infused solutions, represented by the red line, and needle in the left ventricle for the entrance of the perfusion solutions. (**B**) Infusion pump set to move 9 mL/s to deliver phosphate buffered saline (PBS) solution and then paraformaldehyde (PFA). (**C**) The skullcap was removed for exposure and removal of the brain. (**D**) Brain was coronally sliced in a cryostat at 40 µm under a temperature of −24 °C. (**E**) Representative image of the positive TH-stained slices after immunohistochemistry protocol. (**F**) Quantification of the percentage of dopaminergic death in striatum and in substantia nigra pars compacta (SNc) using Image J software.

**Figure 4 nutrients-12-01551-f004:**
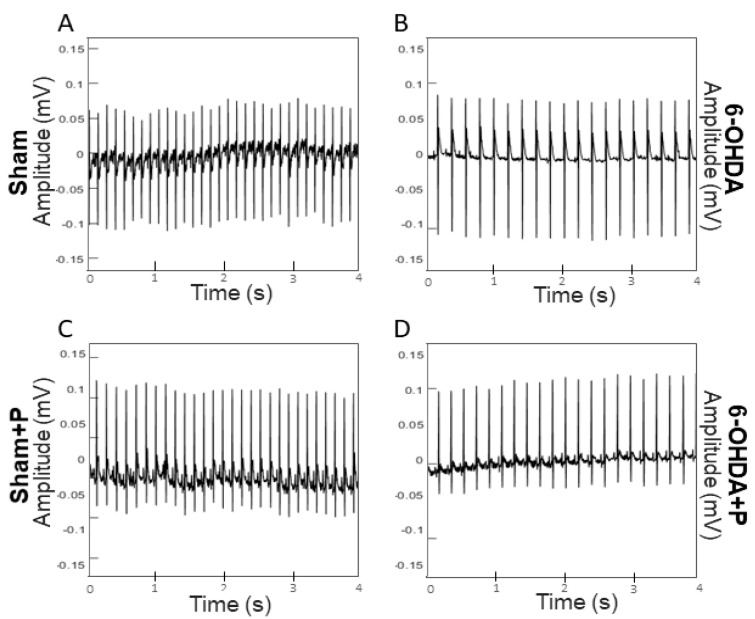
Representative 4 s epoch of the ECG recordings. (**A**) Sham; (**B**) 6-OHDA; (**C**) sham + P; (**D**) 6-OHDA + P.

**Figure 5 nutrients-12-01551-f005:**
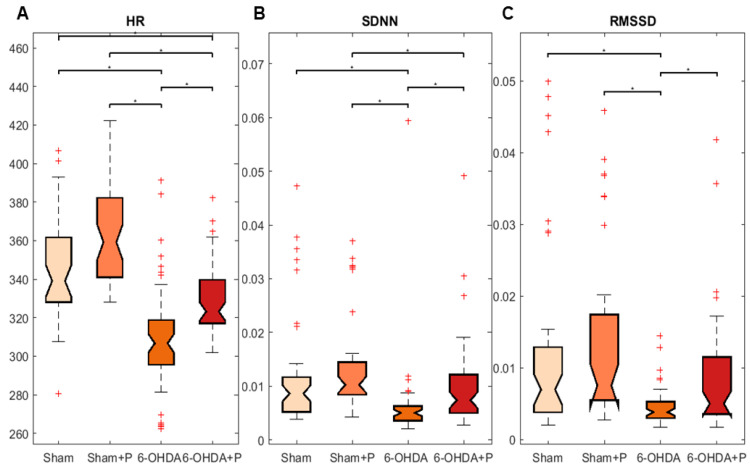
Boxplot of heart rate and heart rate variability parameters (SDNN and RMSS). The interquartile range in the graphs is indicated by the size of the boxes; the confidence interval for median is represented by the chamfer; the statistically significant difference between two groups is indicated by the asterisks; data with values outside the interquartile range are represented by the red ‘+’ symbol. Median and confidence interval for HR, SDNN, and RMSSD were as follows: (**A**) HR: 339.1 ± 8.3 (sham), 359.2 ± 9.3 (sham + P), 306.7 ± 4.8 (6-OHDA), 323.2 ± 4.8 (6-OHDA + P); (**B**) SDNN: 0.008632 ± 0.001508 (sham), 0.01025 ± 0.00154 (sham + P), 0.005022 ± 0.000605 (6-OHDA), 0.007386 ± 0.001517 (6-OHDA + P); (**C**) RMSSD: 0.006963 ± 0.002125 (sham), 0.007532 ± 0.003008 (sham + P), 0.003804 ± 0.000501 (6-OHDA), 0.00499 ± 0.001705 (6-OHDA + P).

**Figure 6 nutrients-12-01551-f006:**
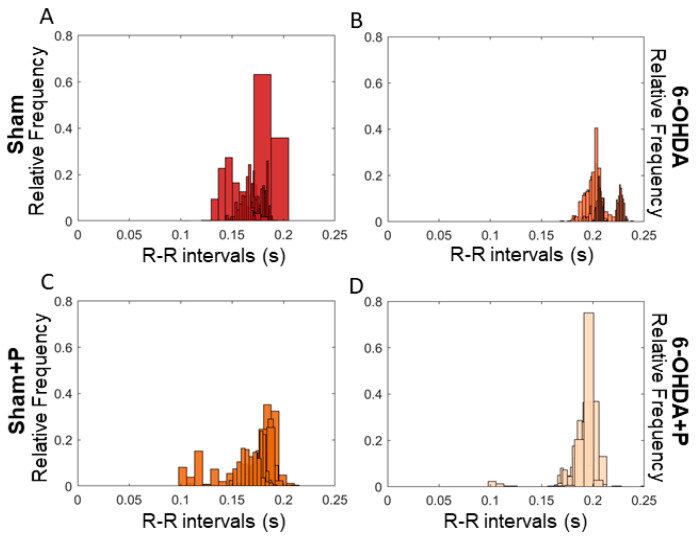
Histograms showing distribution of R-R intervals by length range. (**A**) sham: RR interval from 0.12 to 0.206 s; (**B**) 6-OHDA: RR interval from 0.175 to 0.237 s; (**C**) sham + P: RR interval from 0.095 to 0.212 s; (**D**) 6-OHDA + P: interval RR of 0.15 to 0.212 s.

**Figure 7 nutrients-12-01551-f007:**
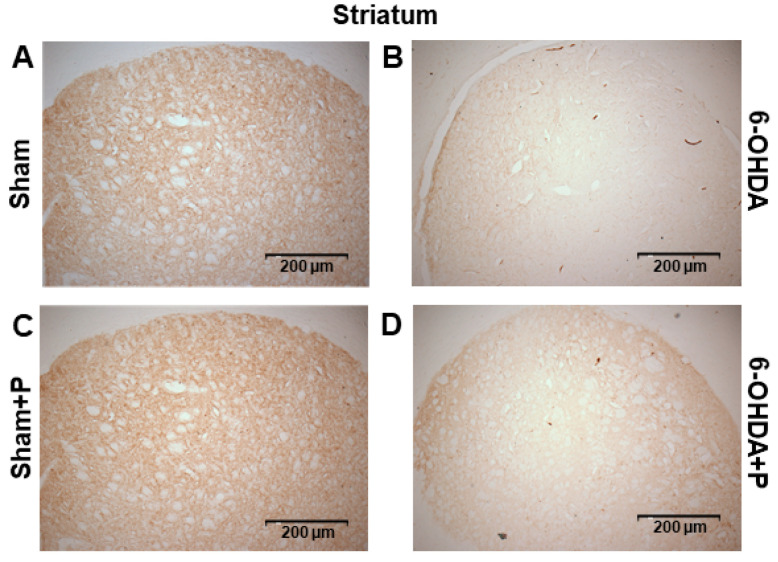
Representative striatum images of tyrosine hydroxylase (TH)-immunoreactive sections. TH-positive striatal fibers in (**A**) sham; (**B**) 6-OHDA; (**C**) sham + P; (**D**) 6-OHDA + P.

**Figure 8 nutrients-12-01551-f008:**
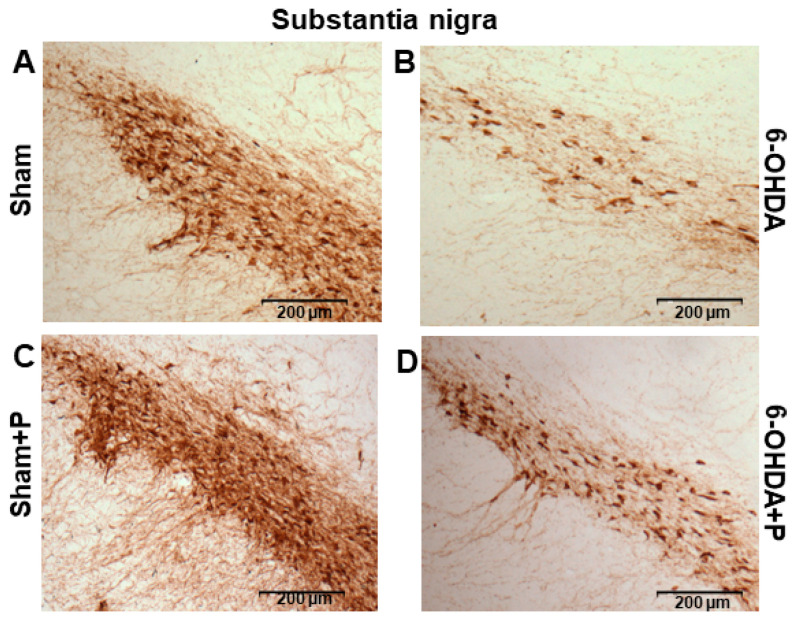
Representative SNc images of TH-immunoreactive sections. TH-positive dopaminergic neurons in the SNc in (**A**) sham; (**B**) 6-OHDA; (**C**) sham + P; (**D**) 6-OHDA + P.

**Figure 9 nutrients-12-01551-f009:**
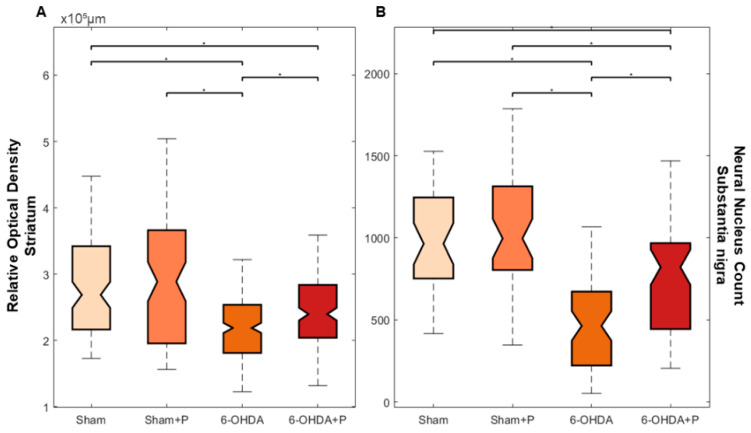
Boxplot of the TH immunohistochemistry. The interquartile range in the graphs is indicated by the size of the boxes; the confidence interval for median is represented by the chamfer; the statistically significant difference between two groups is indicated by the asterisks. (**A**) Optical density of the TH-positive fibers in the striatum; median and confidence interval were as follows: 2.4949 × 10^+5^ ± 0.214 × 10^+5^ (sham); 2.927 × 10^+5^ ± 0.243 × 10^+5^ (sham + P); 2.356 × 10^+5^ ± 0.084 × 10^+5^ (6-OHDA + P); 2.189 × 10^+5^ ± 0.087 × 10^+5^ (6-OHDA). (**B**) TH-positive neuronal nuclei in the SNc; median and confidence interval were as follows: 996 ± 119 (sham); 964 ± 126 (sham + P); 821 ± 166.5 (6-OHDA + P); 463.5 ± 90.6 (6-OHDA).

**Figure 10 nutrients-12-01551-f010:**
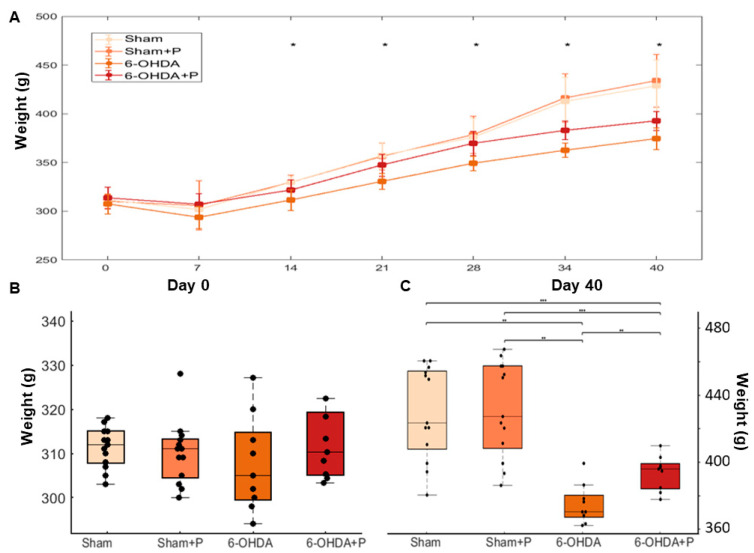
Statistical analysis of evolution of animal weights during the experimental period of 40 days. (**A**) Average weight with standard deviation (groups: sham, *n* = 13; sham + P, *n* = 13; 6-OHDA, *n* = 9; 6-OHDA + P, *n* = 9). The asterisks indicate a significant difference between groups (* *p* < 0.05). Day 0: chi-square = 2.2876, *p* = 0.5149; day 7: chi-square = 5.4888, *p* = 0.1393; day 14: chi-square = 14.8267, *p* = 0.0019709; day 21: chi-square = 17.6751, *p* = 0.00051; day 28: chi-square = 19.057, *p* = 0.000266; day 34: chi-square = 25.7706, *p* = 0.00001065; day 40: chi-square = 25.7568, *p* = 0.0000107. (**B**) Distribution of the animals’ body weight of all groups on the first day of measurement. The mean and standard deviation were as follows: 311.3 ± 4.392 (sham); 310.2 ± 6.9 (sham + P); 307.7 ± 10.2 (6-OHDA); 313.7 ± 11.11 (6-OHDA + P). (**C**) Distribution of the animals’ body weight of all groups on day 40 of measurement. The mean and standard deviation were as follows: 428.8 ± 26.6 (sham); 434.2 ± 27.13 (sham + P); 374.7 ± 11.15 (6-OHDA); 392.8 ± 9.704 (6-OHDA + P).

**Table 1 nutrients-12-01551-t001:** Statistical moments of the distribution of the R-R intervals.

	Mean	Standard Deviation	Skewness	Kurtosis
**Sham**	0.1467	0.0072	1.5494	3.4277
**Sham + Propolis**	0.1723	0.0143	−0.9413	4.9860
**6-OHDA**	0.2415	0.1675	1.0905	3.3249
**6-OHDA + Propolis**	0.1889	0.0080	−0.9608	15.3378

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
