# Peer review of "Propolis as a Potential Disease-Modifying Strategy in Parkinson’s disease: Cardioprotective and Neuroprotective Effects in the 6-OHDA Rat Model"

_nutrients, 2020, doi:10.3390/nu12061551_

Round 1
Reviewer 1 Report
This research is about propolis as a potential disease-modifying strategy in Parkinson’s disease and showed cardioprotective and neuroprotective effects in the 6-OHDA rat model. The manuscript contains interesting findings but needs thorough revision in order to be accepted.
In Introduction, 'Natural compounds are potential candidates for novel adjunctive therapeutic approaches.'. Add example of natural compounds.
Add age of Wistar rats in '2.1. Animals'.
In '2.2. Study Design', 'the animals were euthanized by transcardiac perfusion'. Add the related method (anesthetic..)
In '2.4. Implantation of cardiac electrodes and electrocardiogram', add device name and company.
Add statistics program '2.8. Statistics'.
In Figure 7, can you show the full image of striatum?
Reviewer 2 Report
This is a really excellent presentation of data for a subject of high medical significance for PD patients. They have presented the findings well with rigorous statistical analysis. On first impression, l thought the methodological figures to be unnecessary but on reconsideration they are tremendously helpful for readers who are not experts in the field. They make the paper readable for a wider audience. I am not qualified to comment on the electrophysiology and heart function measurements and data analysis, only the IHC. I learnt a new word "diaphanization"
Minor comments.
1. A couple of typos
2. As the authors are investigating a complex "natural product" with multiple components, could they comment on how variable the difference balances of components might be from batch to batch. At present, it is not known which of the components are most important for the biological effects observed.
